# Neuromuscular Control before and after Independent Walking Onset in Children with Cerebral Palsy

**DOI:** 10.3390/s21082714

**Published:** 2021-04-12

**Authors:** Annike Bekius, Coen S. Zandvoort, Jennifer N. Kerkman, Laura A. van de Pol, R. Jeroen Vermeulen, Jaap Harlaar, Andreas Daffertshofer, Annemieke I. Buizer, Nadia Dominici

**Affiliations:** 1Department of Human Movement Sciences, Faculty of Behavioural and Movement Sciences, Amsterdam Movement Sciences, Institute for Brain and Behavior Amsterdam, Vrije Universiteit Amsterdam, 1081 BT Amsterdam, The Netherlands; a.bekius@vu.nl (A.B.); c.s.zandvoort@vu.nl (C.S.Z.); j.n.kerkman@vu.nl (J.N.K.); a.daffertshofer@vu.nl (A.D.); 2Department of Rehabilitation Medicine, Amsterdam Movement Sciences, Amsterdam UMC, Vrije Universiteit Amsterdam, 1081 HV Amsterdam, The Netherlands; j.harlaar@amsterdamumc.nl (J.H.); ai.buizer@amsterdamumc.nl (A.I.B.); 3Department of Pediatric Neurology, Amsterdam UMC, Vrije Universiteit Amsterdam, 1081 HV Amsterdam, The Netherlands; l.vandepol@amsterdamumc.nl; 4Department of Neurology, School of Mental Health and Neuroscience, Maastricht University Medical Center, 6202 AZ Maastricht, The Netherlands; jeroen.vermeulen@mumc.nl; 5Department of Biomechanical Engineering, Delft University of Technology, 2628 CD Delft, The Netherlands; 6Emma Children’s Hospital, Amsterdam UMC, Vrije Universiteit Amsterdam, University of Amsterdam, 1105 AZ Amsterdam, The Netherlands

**Keywords:** muscle synergies, CP, electromyography, gait, early brain lesions, motor development

## Abstract

Early brain lesions which produce cerebral palsy (CP) may affect the development of walking. It is unclear whether or how neuromuscular control, as evaluated by muscle synergy analysis, differs in young children with CP compared to typically developing (TD) children with the same walking ability, before and after the onset of independent walking. Here we grouped twenty children with (high risk of) CP and twenty TD children (age 6.5–52.4 months) based on their walking ability, supported or independent walking. Muscle synergies were extracted from electromyography data of bilateral leg muscles using non-negative matrix factorization. Number, synergies’ structure and variability accounted for when extracting one (VAF_1_) or two (VAF_2_) synergies were compared between CP and TD. Children in the CP group recruited fewer synergies with higher VAF_1_ and VAF_2_ compared to TD children in the supported and independent walking group. The most affected side in children with asymmetric CP walking independently recruited fewer synergies with higher VAF_1_ compared to the least affected side. Our findings suggest that early brain lesions result in early alterations of neuromuscular control, specific for the most affected side in asymmetric CP.

## 1. Introduction

Cerebral palsy (CP) is a neurodevelopmental disorder caused by brain lesions before birth or early in life [1,2]. It covers a wide clinical spectrum, from children who manage to walk independently, to children being completely wheelchair dependent. Children can be affected symmetrically (bilateral CP) or asymmetrically (unilateral CP or asymmetric bilateral CP) [3]. Topography and severity of CP can be difficult to predict in infancy [4].

Typically, infants take their first independent steps between the age of 9 to 18 months, representing an important milestone in motor development [5]. Reaching this milestone can be challenging for children with cerebral palsy (CP). Early interventions can be critical to improve motor functions, including walking, because the neural networks under development are still highly plastic [6]. To improve interventions that promote functional mobility in children with CP, it is important to identify neuromuscular mechanisms of abnormal motor development as early as possible.

One possibility to assess these neuromuscular mechanisms is the use of muscle synergy analysis. The central nervous system has been theorized to reduce the degrees of freedom in the coordination of muscle activation during walking through basic building blocks, named muscle synergies or locomotor modules, that resemble identical temporal activation patterns of groups of muscles [7,8,9,10]. The number of basic activation patterns in typically developing (TD) children increases from two during neonate stepping, to four in toddlers who have just started to walk independently [7]. Muscle synergy analysis has recently been adopted to quantify neuromuscular control during walking in school-age children with CP, and has been shown to provide a consistent measure between days [11].

Despite a limited number of recorded muscles previous studies show that older children with CP recruit fewer synergies during walking compared to age-matched TD children [12,13,14,15,16,17]. In addition, several studies reported that the walking patterns of older children with CP retain some of the characteristics of the younger TD children, by showing the excessive muscular co-contraction of only a few muscles [18,19,20]. The small number of recorded muscles and the age of the children involved in these studies limits our current understanding of neuromuscular control in very young children with early brain lesions [6,21,22]. A more detailed and comprehensive assessment of multi-muscle coordinate patterns is needed [23].

The time before the onset of independent walking can be a critical period for early interventions to improve motor functions including walking [6,24]. Previous studies compared children with CP and TD children of similar age, while it may be relevant to match these groups for developmental phase [17]. One study from Prosser et al. 2010 [25] compared the trunk and hip muscles in children with CP and TD with similar walking experience (an average of 28.5 months of walking experience). Nevertheless, the age range of the children with CP included in this study was quite large (2 to 9 years old), as was the walking experience (0.5 to 60 months). In addition, they did not differentiate between children able to walk independently and children who needed support, e.g., by using an assistive device, to perform the walking task.

The aim of this study was to assess whether neuromuscular control in young children with CP differs from that of TD children with the same walking ability in the early phase of motor development, i.e., before the onset of independent walking (supported walking) and just after the onset of independent walking in particular by means of the contributing muscle synergies. In addition, we examined whether there was a difference in neuromuscular control between the most and least affected side of children who were affected asymmetrically (unilateral or asymmetric bilateral CP). We hypothesized that already before or during the first years of independent walking, children with (high risk of) CP recruit fewer synergies compared to TD children and that this is specific to the most affected side in children with asymmetric CP.

## 2. Materials and Methods

### 2.1. Participants

Children with early brain lesions, at high risk of CP or with an established diagnosis of CP (referred to as CP group) were recruited from the Departments of Pediatric Rehabilitation and of Child Neurology at Amsterdam University Medical Centers (Amsterdam UMC), and from the Department of Child Neurology at Maastricht University Medical Center (Maastricht UMC+); see Table 1 for the in- and exclusion criteria. TD children (referred to as TD group) were recruited by word of mouth. Participants were in either the supported walking (SW) or independent walking (IW) group, based on their walking ability. Children in the SW group could not walk independently, while children in the IW group could. Individual and average characteristics of both groups, as well as the clinical characteristics of the children in the CP group, are listed in Table 2.

The study was approved by the Ethics Committee of the Faculty of Behavioral and Movement Sciences at the Vrije Universiteit Amsterdam (VCWE-2016-082) for the TD group, and of Amsterdam UMC (NL59589.029.16) for the CP group. In Maastricht UMC+, local practicability was granted. The parents of all children were informed about the procedure of the study and provided written informed consent prior to participation in accordance with the declaration of Helsinki for medical research involving human participants.

### 2.2. Procedure

Experiments were performed in the clinical gait laboratories of the Department of Rehabilitation Medicine at Amsterdam UMC (location VUmc) and Maastricht UMC+, and the BabyGaitLab laboratory of the Department of Human Movement Sciences at the Vrije Universiteit Amsterdam. The responsible investigators and one or both parents of the child were present during the experiments, and, for the CP group, also a pediatric physio-therapist. At the start of each experiment for the CP group, the pediatric physiotherapist performed a physical examination, to identify possible motor asymmetry in the CP group (in this case reported as asymmetrically affected child).

Children in the SW group walked with support on a treadmill (with the exception of one child who walked over-ground). Adequate support was provided by the physiotherapist, experimenter or parent that held the trunk of the child with both hands or held the hand of the child, while the other parent, or an experimenter, encouraged the child to take steps [32,33]. The treadmill speed was adjusted to induce a walking pattern and tuned to a comfortable speed for the child. Children in the IW group walked independently over-ground, and were encouraged to walk in a straight line, at their preferred walking speed. Only sequences of steps executed naturally by the child were considered.

Muscle activity was recorded with surface electromyography (EMG) from 18 to 22 bilateral leg- and trunk muscles simultaneously using Mini Wave wireless EMG systems (Cometa, Italy). The following muscles were recorded from each side: tibialis anterior (TA); gastrocnemius medialis (GM); gastrocnemius lateralis (GL); soleus (SOL); rectus femoris (RF); vastus medialis (VM); vastus lateralis (VL); biceps femoris (BF); tensor fascia latae (TFL); gluteus maximus (GLM) and erector spinae (ES) at L2 level. EMG electrode placement was performed according to the Surface Electromyography for the Non-Invasive Assessment of Muscles protocol [34], and the standard recommendations for minimizing cross-talk between adjacent muscles [7,10,35]. The skin was cleaned with alcohol and mini golden reusable surface EMG disc-electrode pairs (15 mm diameter, acquisition area 4 mm^2^) were placed at the approximate location of the muscle. To minimize movement artefacts, pre-amplified EMG sensor units were attached to the skin of the child and fixed with elastic gauzes. The signals were amplified and sampled at 1000 Hz. Body kinematics and high-speed video were recorded at 100 Hz using a VICON system (Oxford, UK). 32-channel electro-encephalography (EEG) recordings were performed, but not analyzed here. Sampling of EMG, video, and kinematic data was synchronized online.

### 2.3. Spatiotemporal Gait Parameters

The step events were extracted from the video and confirmed with kinematic data for both sides. The gait cycle was defined as a cyclic movement of one leg, starting when the foot strikes the ground and ending when the foot of the same leg strikes again. The end of stance was defined as the moment when the foot lifts off the ground. Gait initiation/termination strides and jumps or turning were discarded from analysis. Stride velocity was calculated using the corresponding stride length and stride duration. The stride length was computed according to the 3D displacement of the foot marker. For the trials recorded during walking on the treadmill, the treadmill speed was taken into consideration to correct for the participants’ displacements. Stance duration, i.e., from foot strike to foot off, for both legs was computed.

### 2.4. Muscle Synergy

All analyses were conducted in MATLAB (version 2017b, Mathworks Inc., Natick, MA, USA). Raw EMG data were processed offline according to the following sequence: Notch filter (50 Hz), high pass filtering (30 Hz), full-wave rectification, and low pass filter with a fourth order Butterworth filter (10 Hz).

Muscle synergies were extracted using non-negative matrix factorization (NMF) of the pre-processed EMG data [36]. EMG amplitudes of each muscle were normalized to the maximum of the mean value across all strides plus its standard deviation (SD) for each participant and the timescale was normalized to t=201 data points per gait cycle for each limb. Briefly, the NMF was applied to the mean EMG envelopes for each participant, and decomposed the EMG data into temporal activation patterns (P) and synergy weights (W), according to the following equation:(1)$$EMG\;=\overset{n}{\underset{i=1}{\sum }}P_{i}W_{i}+e\;,$$
where the pre-processed EMG data (m×t matrix, where m is the number of muscles and t is the number of time points) is a linear combination of the temporal activation patterns P (n×t matrix, where n≤m is a predetermined number of synergies) and synergy weights W (m×n matrix), and e denotes the residual error.

A set of 1–8 synergies was extracted with a restriction of 100 maximum iterations, 1000 replicates, and a threshold for convergence and completion of 10^−4^. NMF was applied to the EMG activity of bilateral muscles (including both sides), and unilateral muscles (including one side). The results of the unilateral EMG analysis were used separately to compare the most and least affected side in children with asymmetric CP, and right and left side in children with symmetric CP and TD children.

The reconstruction accuracy of the extracted synergies was determined by the variability accounted for (VAF), which is the ratio of the sum of squared errors to the total sum of squares computed with respect to the mean [7,37,38]. Next to VAF we also determined the synergies’ contribution to the matrix (or Frobenius) norm [39,40,41], which revealed comparable results. For the sake of legibility, we here present the conventionally used VAF. The minimum number of synergies to approximate the pre-processed EMG was defined as required for VAF to exceed 85%, or when the added VAF of the following synergy was below 8% [42,43,44]. In addition, the selected number of synergies had to account for more than 80% VAF for every individual muscle [38,45,46]. Since setting of a VAF threshold arguably comes with arbitrariness, we also investigated VAF by one synergy (VAF_1_) in the unilateral EMG analysis, previously used as a summary measure of synergy complexity related to function and treatment outcome [14,47], and by two synergies (VAF_2_) in the bilateral EMG analysis [7,48]. For the latter, we included an additional synergy to account for the mirror muscle activations in the contralateral side, shifted by 50% of the gait cycle.

In the bilateral EMG analysis, number of synergies and VAF_2_ were compared between CP and TD for the SW and IW group. To compare temporal activation patterns and synergy weights between groups without a restriction to a certain threshold, the number of synergies was fixed to four, which is the number of synergies typically reported in healthy adults during walking [7,46,49]. The patterns were grouped and plotted according to the timing of the main peak relative to the normalized gait cycle. Average temporal activation patterns and synergy weights per group for each synergy were compared between CP and TD for the SW and IW group.

In the unilateral EMG analysis, number of synergies and VAF_1_ for the most affected side in the asymmetric CP group, or a random side in the symmetric CP and TD group (random side), were compared between CP and TD for the SW and IW group. Furthermore, the number of synergies and VAF_1_ were compared between most and least affected side in the asymmetric CP, and right and left side in symmetric CP and TD group.

### 2.5. Statistical Analysis

All data are reported as mean ± SD. An independent *t*-test was used when the data was normally distributed, and a one-tailed non-parametric Mann-Whitney U-test when it was not. An independent *t*-test in statistical parametric mapping was performed to assess the similarity of temporal activation patterns per synergy between CP and TD for the SW and IW group. Synergy weights were compared between groups using Pearson’s correlation coefficients, where r > 0.7 represented high similarity and r > 0.45 marginal similarity [50]. Significance threshold was set at *p* < 0.05 for all tests. For the comparison between unilateral results within groups, the Wilcoxon signed rank test was used for number of synergies, and a paired samples *t*-test for VAF_1_.
sensors-21-02714-t002_Table 2Table 2Participant characteristics.ParticipantGenderAge(mo)CA (mo)WO (mo)Distribution Subtype GMFCSScores Brain Damage (Side)BW (kg)N StridesSpeed (km/h)CP1M11.610.6-Uni RspasticNSb2 (bi)9.7170.39CP2F14.815.1-Uni LspasticNS5 (bi)10.8240.60CP3M21.021.4-Uni LspasticNS6 (uni R)9.3290.63CP4F17.817.9-Bi (L > R)spasticNSb2 (bi)10.7450.82CP5F20.217.2-Bi (R > L)spasticNS5 (uni L)7.7310.60CP6M6.56.6-BispasticNSb2 (bi)7.3220.64CP7M9.86.5-BispasticNS4 (bi) -270.80CP8F8.58.9-BiundefNSb2 (bi)8.8430.61CP9F42.841.2-BispasticIII4 (bi)14310.80CP10F44.943.7-BispasticII4 (bi)11.4400.62**CP SW**6 F; 4 M 19.8 (13.6)18.9 (13.4) ^#^-----10.0 (2.1)31 (9)0.65 (0.13)CP11M23.822.217.1Uni RspasticI5 (uni L)11.1351.77CP12M35.635.816.1Uni RspasticIb1 (uni L)13.4272.62CP13M41.038.016.0Uni RspasticI5 (uni L)14.6402.33CP14M47.245.515.6Uni RspasticI5 (uni L)15.2453.87CP15F22.322.315.0Bi (L > R)spasticI2 (bi)10.6421.66CP16M27.826.919.1Bi (R > L)spasticI4 (bi)14.1662.75CP17M38.638.916.1Bi (R > L)spasticIb2 (bi)14.0414.01CP18M18.318.615.0BispasticI4 (bi)10.9184.00CP19F34.429.924.4BiataxicIIb210.0592.37CP20M34.429.926.7BispasticII4 (bi)11.8272.86**CP IW**2 F; 8 M32.3 (9.1)30.8 (8.6)18.1 (4.1) *----12.6 (1.9)40 (15)2.82 (0.87)TD1F6.36.2-----6.8170.41TD2F7.57.8-----9.1330.44TD3M9.710.2-----9.7800.55TD4M9.89.7-----8.5590.60TD5F10.010.0----**-**8.9950.54TD6M10.210.1-----10.2790.69TD7F10.410.2-----9.3660.61TD8F10.69.7-----9.0230.90TD9F11.211.6-----9.6430.66TD10M12.012.0-----11.0210.46**TD SW**6 F; 4 M 9.8 (1.7) ^#^9.8 (1.7)-----9.2 (1.1)52 (28)0.59 (0.14)TD11M16.516.510.7----11.3272.40TD12F17.517.811.6----10.7382.76TD13F19.319.312.9-----931.68TD14F19.719.612.9----10.4492.99TD15F20.120.113.9----10.3862.48TD16M20.820.414.9----13.0663.35TD17F24.424.311.7----11.3451.94TD18M27.527.311.3----13.0213.16TD19F47.147.214.3---**-**16.0493.27TD20M53.552.411.3----15.5283.47**TD IW**6 F; 4 M26.6 (12.9)26.5 (12.7)12.6 (1.4) *----12.4 (2.2)50 (25)2.76 (0.63)Distribution is based on the physical examination performed by a pediatric physiotherapist during the recording. Asymmetrically affected children in the CP group are highlighted in grey. The brain damage scores are defined according to a semi-quantitative MRI scale [51]: 2, full-term border-zone infarction; 4, periventricular leukomalacia; 5, posthemorrhagic porencephaly/venous infarction; 6, middle cerebral artery infarction; b1, developmental brain malformations; b2, non-specific lesions. The mean (SD) is reported for age, corrected age, walking onset (for the IW groups), body weight, number of strides and walking speed. ^#^ indicates a significant difference in age between CP and TD in the SW group (*p* = 0.047), and * indicates a significant difference in age at independent walking onset between CP and TD in the IW group (*p* < 0.001). CP = cerebral palsy; TD = typically developing; SW = supported walking; IW = independent walking; F = female; M = male; CA = corrected age; WO = corrected age at independent walking onset; Bi = bilateral; Uni = unilateral; L = left; R = right; GMFCS = gross motor function classification system; NS = not yet specified; BW = body weight; kg = kilograms; N = number; SD = standard deviation.


## 3. Results

Twenty children in the CP group (corrected age 6.5–45.5 months) and twenty children in the TD group (age 6.3–53.5 months) participated in this study, with ten children in the SW and IW groups (Table 2). Based on the physical evaluation performed in the children of the CP group at the start of each experiment by an expert pediatric physiotherapist we identified a total of *n* = 5 and *n* = 7 asymmetrically affected children in the SW and IW group. Time since onset of independent walking did not significantly differ between the CP and TD group (12.7 ± 9.8 vs. 13.9 ± 12.8 months; *p* = 0.81).

### 3.1. Spatiotemporal Gait Parameters

Stride duration did not differ between CP and TD for SW (2.0 ± 0.5 vs. 2.1 ± 0.6 s) and IW (0.8 ± 0.1 vs. 0.8 ± 0.1 s). Relative stance duration in the IW group was significantly longer for TD (62 ± 4%) compared to CP (58 ± 3%; *p* = 0.02), but CP (72 ± 6%) and TD (71 ± 4%) did not differ significantly in the SW group (*p* = 0.29). Stride velocity did not differ significantly between CP and TD for SW (0.7 ± 0.1 vs. 0.6 ± 0.1 km/h; *p* = 0.15) and IW (2.8 ± 0.9 vs. 2.7 ± 0.7 km/h; *p* = 0.43). Stance duration of the most affected leg of children with asymmetric CP in the IW group was significantly shorter compared to the least affected leg (57 ± 3 vs. 60 ± 4%; *p* = 0.03), while this was not the case when comparing legs in children with symmetric CP (57 ± 5 vs. 58 ± 4%), and TD children (62 ± 3 vs. 62 ± 4%; *p* = 0.44).

### 3.2. Muscle Synergy

In the SW group, bilateral EMG analysis revealed that children in the CP group recruited two, three, or four muscle synergies, and in the TD group three or four synergies. In the IW group, children in the CP group recruited either three or four synergies, and in the TD group three, four, or five synergies (Figure 1A,B and Appendix A). The mean number of synergies that explained the variability in the EMG data was lower in CP compared to TD for SW (2.9 ± 0.7 vs. 3.2 ± 0.4; *p* = 0.14) and IW (3.7 ± 0.5 vs. 4.1 ± 0.6; *p* = 0.06), albeit not reaching statistical significance. VAF2 was higher in CP compared to TD for SW (70.5 ± 11.9 vs. 61.4 ± 7.4%; *p* = 0.03) and IW (60.8 ± 8.2 vs. 53.4 ± 11.0%; *p* = 0.05, Figure 1C). When the number of synergies was fixed to four, we did observe a significant difference between CP and TD in the temporal activation pattern P2 during early swing in the SW group (75.9 to 78.1% of the gait cycle; *p* = 0.04), and P1 during mid swing in the IW group (71.4 to 76.6% of the gait cycle; *p* = 0.01). Correlations between mean synergy weights of CP and TD were high (r > 0.7) for all synergies in the IW group, whereas in the SW group only W4 showed a high and W2 a moderate (r > 0.45) correlation (Figure 2).

The unilateral EMG analysis showed that the mean number of muscle synergies that explained the variability in the EMG data was significantly lower in CP compared to TD for SW (2.7 ± 0.5 vs. 3.1 ± 0.3; *p* = 0.02) and IW (3.0 ± 0.5 vs. 3.6 ± 0.5; *p* < 0.01). VAF1 was significantly higher in CP compared to TD for SW (42.0 ± 12.9 vs. 22.0 ± 10.0%; *p* < 0.001) and IW (39.4 ± 13.3 vs. 29.5 ± 9.5%; *p* = 0.03; Figure 3A).

When comparing the most and least affected side in asymmetric CP, and right vs. left side in symmetric CP and TD children, we did not find any significant differences for the number of synergies and VAF1 in the SW group (Figure 3B,C). However, in the IW group the mean number of synergies in asymmetric CP (*n* = 7) was significantly lower (3.0 ± 0.2 vs. 3.7 ± 0.2%; *p* = 0.01) and VAF1 was significantly higher (36.8 ± 13.7 vs. 28.0 ± 9.2%; *p* < 0.01) for the most affected compared to the least affected side. No significant differences between sides could be identified in TD children (*n* = 10). Due to a limited number of children in the symmetric CP group (*n* = 3) no statistical comparison was performed (Figure 3C).

## 4. Discussion

Our study shows that children with CP recruit fewer muscle synergies compared to TD children already in the supported walking phase, and in the first years after onset of independent walking. To our knowledge, this study is the first to investigate neuromuscular control using muscle synergy analysis during supported and independent walking in such a young patient population. Our results are consistent with the hypothesis that the complexity of neuromuscular control is already reduced in very young children with CP. The critical developmental window may be before the age of two years old, when the brain is highly plastic and the corticospinal tract is still maturing [6,24,52]. The novelty of our findings is that the difference in neuromuscular control of walking is present in children with CP compared to TD children with the same level of walking ability during this early phase of motor development.

Unilateral EMG analysis revealed that children in the CP group recruited fewer synergies during walking, and VAF_1_ was higher compared to the TD group before (during SW) and after the onset of independent walking (during IW), which is in line with previous results on VAF_1_ obtained in older children [14,53,54]. In these studies, however, no distinction was made between children with CP who used an assistive device to walk, and children who walked independently. In addition, the children with CP were compared with TD children of the same age, who very often had more experience of walking independently. Especially since the first independent steps, in children with CP, are typically delayed or will not occur at all. Using the walking experience, and not age, to compare motor ability in children with CP and TD children is an innovative approach. This allows for a more reasonable comparison, and to control for the improvement in walking pattern that occurs after the onset of independent walking.

In children with asymmetric CP walking independently, the number of synergies was lower, and VAF_1_ higher, in the most affected compared to the least affected side. This suggests that ‘simplified’ neuromuscular control is specific to the most affected side in children who just start to walk independently. In bilateral EMG analysis, the least affected side in children with asymmetric CP walking independently may level out the effect of the most affected side, which can explain the lack of statistically significant differences in number of synergies between CP and TD, at least to some degree. The difference between the most and least affected side was not present in the asymmetric CP group during SW, which may have been caused by the support the children received. Alternatively, it might imply that this difference between sides occurs later in motor development.

When we removed the restriction of a threshold by fixing the number of synergies to four, we found only minor significant differences in temporal activation patterns, and synergy weights were highly correlated during IW between the CP and TD group. This confirms previous studies that reported comparable synergy structures during walking in children with CP with high functional mobility levels and TD children [37,49]. The lower number of synergies in children with CP walking independently, as defined by a VAF threshold, possibly results from the merging of the four synergy structures [46,55]. If true, this means that these children may have access to four synergies, but are not able to recruit all synergies independently. During SW, a fourth synergy did not contain additional information, since all muscles were generally active for both CP and TD, showing that on average these children could not access four synergies.

The variability in results from the EMG analysis between individuals was high. This may reflect the heterogeneity of the CP group, including children with various brain lesions and functional mobility levels, as was emphasized in previous studies [12,14,17]. The population investigated is very young, and although we compared between children in the same developmental stage variability between children within groups may be large. Despite variability between individuals and within relatively small groups, we found significant differences between the CP and TD group during SW and IW.

Some limitations of this study have to be recognized. In young children at high risk of CP, we did not always know whether they would actually develop CP. Not all children with a high risk of CP eventually receive a diagnosis of CP, and motor types and topography may emerge and change during the first two years of life [4]. Another limitation of the study was that a relatively small number of participants was included in the IW symmetric CP group, and as a consequence, we could not perform a statistical comparison between sides. Walking speed is an important factor to consider in muscle synergy analysis, since some studies found that walking speed affected the number and structure of muscle synergies [14,56,57]. The children in our study walked at their preferred and comfortable speed, which may have caused variability within the groups. However, average walking speed did not significantly differ between the CP and TD group during both SW and IW, and thus walking speed did not influence the results of the group comparison.

Our results encourage further investigation of the use of muscle synergy analysis as an objective tool for early detection of impaired neuromuscular control. This can help to identify candidates for targeted early interventions aimed at improving neuromuscular control and walking development. Future research should investigate the longitudinal development of muscle synergies within children during development from supported walking to independent walking to minimize the inter-subject variability.

## 5. Conclusions

In conclusion, our study shows that young children with CP, or at high risk of CP, recruit fewer synergies compared to TD children with the same walking ability already in the early phase of motor development. The most affected side in children with asymmetric CP walking independently employed fewer synergies than the least affected side. This suggests that brain lesions in CP result in early alterations of neuromuscular control.

## Figures and Tables

**Figure 1 sensors-21-02714-f001:**
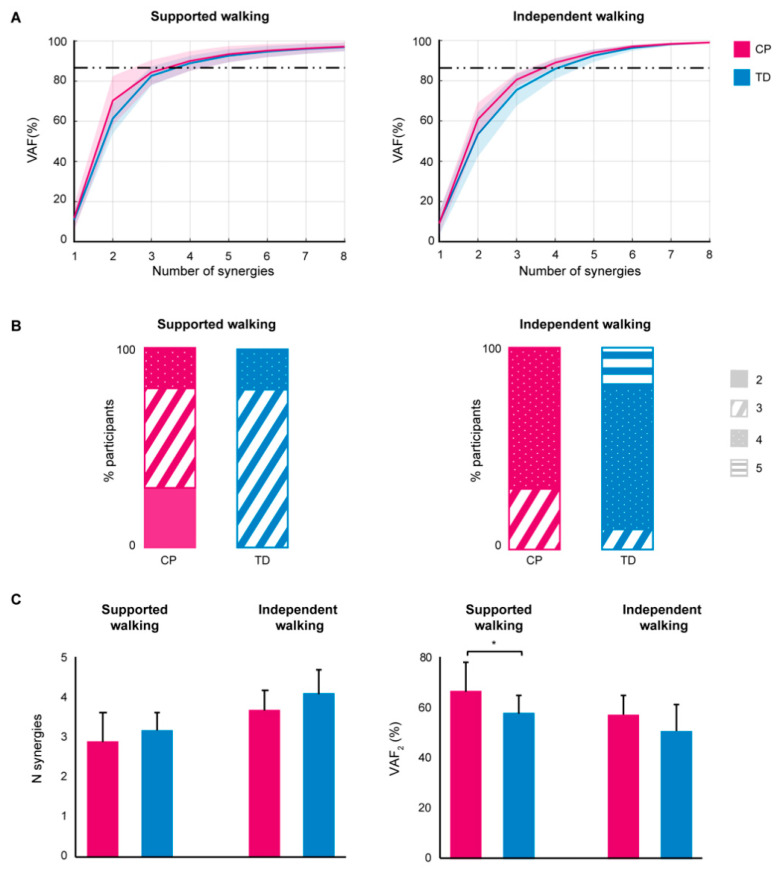
Bilateral EMG analysis results. (**A**) Mean variability accounted for (VAF) ± SD, per synergy 1–8 for the cerebral palsy (CP) and typically developing (TD) group, for supported walking (*left*) and independent walking (*right*); (**B**) Percentage number of synergies per group (*n* = 10) based on a VAF threshold of 85% or added VAF < 8% for the CP and TD group, for supported (*left*) and independent walking (*right*); (**C**) Mean number (N) of synergies (*left*) and variability accounted for (*right*) by two synergies (VAF_2_) for the CP and TD group, for supported walking and independent walking. Error bars indicate SDs, * *p* < 0.05.

**Figure 2 sensors-21-02714-f002:**
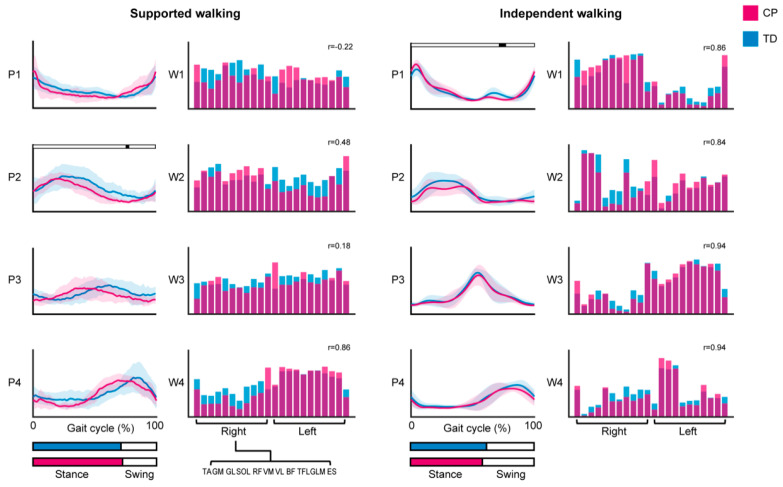
Mean activation patterns and synergy weights for a fixed number of four synergies. Bilateral EMG analysis results of the cerebral palsy (CP) and typically developing (TD) group, for supported walking (*left*) and independent walking (*right*). Lines show the mean temporal activation patterns (P) ± SD along the gait cycle, with mean stance and swing phase indicated by bar graphs at the bottom. Synergy weights (W) for the recorded muscles are depicted in a bar graph. Significant differences between activation patterns are indicated by the black bars, *p* < 0.05. Pearson’s correlations coefficients (r) between mean synergy weights of the CP and TD group are given. Abbreviations: TA = tibialis anterior; GM = gastrocnemius medialis; GL = gastrocnemius lateralis; SOL = soleus; RF = rectus femoris; VM = vastus medialis; VL = vastus lateralis; BF = biceps femoris; TFL = tensor fascia latae; GLM = gluteus maximus; ES = erector spinae at L2 level.

**Figure 3 sensors-21-02714-f003:**
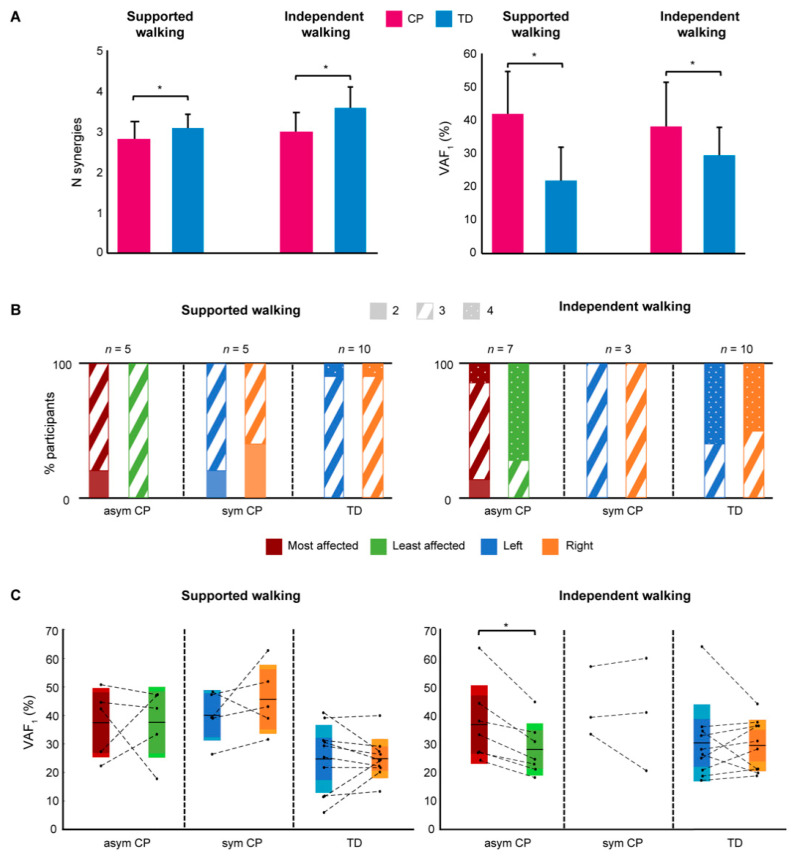
Unilateral EMG analysis results. (**A**) Mean number (N) of synergies (*left*) and variability accounted for (*right*) by one synergy (VAF1) for the CP and TD group, for supported walking and independent walking. Error bars indicate SDs. The unilateral EMG analysis involves the most affected side for the asymmetric CP group and a random side for the symmetric CP and TD group. (**B**) Percentage number of synergies for each side. The most affected (red) and least affected (green) side for asymmetric CP, and the left (blue) and right (orange) side for symmetric CP and TD. (**C**) Variability accounted for by one synergy (VAF_1_) for each side. Mean VAF_1_ per side is indicated by the black line in the middle of the box. The dark color of the box indicates the 95% confidence interval, and the lighter color one standard deviation. Individual participant values are indicated by black dots, and both sides of each participant are connected by broken lines. Significance between sides is indicated by a *, *p* < 0.05. Abbreviations: asym = asymmetric; sym = symmetric; *n* = number of participants.

**Table 1 sensors-21-02714-t001:** In- and exclusion criteria for the CP group.

Inclusion Criteria	Exclusion Criteria	
Diagnosis of CP based on the predominant type of motor impairment and classified according to the criteria proposed by Himmelmann et al. (2005) [26]. CP diagnosis was confirmed according to medical history, brain magnetic resonance results and clinical examination, OR, in children under 24 months: At high risk for developing CP, based on the presence of one of the following [27,28]: - Cystic periventricular leukomalacia, diagnosed on serial ultrasound assessments of the brain [29] - Unilateral or bilateral parenchymal lesion of the brain, diagnosed using MRI [30] - Term/near-term asphyxia resulting in Sarnat 2 or 3 [31] with brain lesions on MRI and/or with neurological dysfunction during infancy suggesting the development of CP - Neurological dysfunction suggestive of development of CP	Functional surgery on bones and/or muscles of the legs Selective dorsal rhizotomy in the last 12 months Severe epilepsy GMFCS IV and V Above the age of five years Brain damage above the age of one year	

Abbreviations: CP = cerebral palsy; GMFCS = gross motor function classification system; MRI = magnetic resonance imaging.

## Data Availability

The data that support the findings of this study are available from the authors A.B. and N.D., upon reasonable request.

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
