# Peer review of "Neuromuscular Control before and after Independent Walking Onset in Children with Cerebral Palsy"

_sensors, 2021, doi:10.3390/s21082714_

Round 1

Reviewer 1 Report

In this paper, to clarify the differences between the neuromuscular control of young children with cerebral palsy (CP) and that of typically developing (TD) children, the muscle synergies are analyzed by using surface electromyography (EMG) data. As a result, this paper shows that young children with CP obtain fewer synergies compared to TD children.

The experiments are well organized. Moreover, the analytical methods and their results are also well described. Therefore, this paper is specialistic, but interesting for many readers.

Reviewer 2 Report

The authors propose a study about neuromuscolar before and after independent walking onset in children.

The proposed study is interesting but there are some points that the authors should better discuss.

The authors should be better described the novelties of their study with respect to existing ones. Furthermore, the authors should provide more details and discussion about the obtained results. The Conclusion section also needs to be improved by discussing practical implications of their analysis.

I suggest to further analyze more recent approaches about the examined topics. In particular, I suggest the following papers to further investigate psychological aspects of examined partecipants by using deep learning models:

1) An emotional recommender system for music. IEEE Intelligent Systems.

Finally, I suggest to perform a linguistic revision.

Reviewer 3 Report

Early brain lesions which produce cerebral palsy (CP) may affect the development of walking. The authors here they grouped twenty children with (high risk of) CP and twenty TD children (age 6.5-52.4 months) based on their walking ability, supported or independent walking. The authors are  extracted the  muscle synergies were extracted from electromyography data of bilateral leg muscles using non-negative matrix factorization. Their findings suggest that early brain lesions result in early alterations of neuromuscular control, specific for the most affected side in asymmetric CP.

Significant: No, the paper is not a significant advance or contribution.

Supported: Mostly yes,

Referencing: some additions are necessary. 

Quality: The organization of the manuscript and presentation of the data and results need some improvement.

Data: Yes, but some results are necessary

Whilst the paper shows promising initial results. A key theme includes the need to focus on the novel contributions and limit the explanation of well-known methods and definitions in the field. The paper is not clear. I also recommend major reorganisation of the paper.

Round 2

Reviewer 2 Report

I think that the authors have addressed all my concerns.

Reviewer 3 Report

The Authors responded adequately to all my comments/suggestions . The paper has been improved, and the quality of the work has improved a lot.